# Spatial-Temporal Pattern Evolution of Public Sentiment Responses to the COVID-19 Pandemic in Small Cities of China: A Case Study Based on Social Media Data Analysis

**DOI:** 10.3390/ijerph191811306

**Published:** 2022-09-08

**Authors:** Yuye Zhou, Jiangang Xu, Maosen Yin, Jun Zeng, Haolin Ming, Yiwen Wang

**Affiliations:** School of Architecture and Urban Planning, Nanjing University, Nanjing 210093, China

**Keywords:** spatial-temporal pattern evolution, public sentiment, COVID-19 pandemic, mental health resilience, social media data

## Abstract

The impact of the COVID-19 pandemic on public mental health has become increasingly prominent. Therefore, it is of great value to study the spatial-temporal characteristics of public sentiment responses to COVID-19 exposure to improve urban anti-pandemic decision-making and public health resilience. However, the majority of recent studies have focused on the macro scale or large cities, and there is a relative lack of adequate research on the small-city scale in China. To address this lack of research, we conducted a case study of Shaoxing city, proposed a spatial-based pandemic-cognition-sentiment (PCS) conceptual model, and collected microblog check-in data and information on the spatial-temporal trajectory of cases before and after a wave of the COVID-19 pandemic. The natural language algorithm of dictionary-based sentiment analysis (DSA) was used to calculate public sentiment strength. Additionally, local Moran’s I, kernel-density analysis, Getis-Ord Gi* and standard deviation ellipse methods were applied to analyze the nonlinear evolution and clustering characteristics of public sentiment spatial-temporal patterns at the small-city scale concerning the pandemic. The results reveal that (1) the characteristics of pandemic spread show contagion diffusion at the micro level and hierarchical diffusion at the macro level, (2) the pandemic has a depressive effect on public sentiment in the center of the outbreak, and (3) the pandemic has a nonlinear gradient negative impact on mood in the surrounding areas. These findings could help propose targeted pandemic prevention policies applying spatial intervention to improve residents’ mental health resilience in response to future pandemics.

## 1. Introduction

COVID-19 has become a global public health emergency, with continuous outbreaks worldwide bringing massive health disasters to human society [1]. It spreads more rapidly and is more destructive in cities and towns with a high population density, which is not conducive to the construction and development of healthy cities. According to a report released by the World Health Organization (WHO) [2] in March 2022, the global pandemic of COVID-19 will not only cause damage to people’s physical health but also increase the mental stress of people everywhere. In the first year of the outbreak alone, the global prevalence of anxiety and depression increased dramatically, by 25%, with young people particularly affected. The pandemic itself and the social isolation it entails may trigger fear, sadness, anxiety, depression and other negative emotions, affecting people’s mental health [3,4,5]. If increasing negative public opinions are not found and treated in time, they may affect social stability. What is worse, they are not conducive to the implementation of effective measures to control the pandemic. In general, public health systems, the media and the public focus more on the consequences of the pandemic, while mental health issues that co-occur with the disease are somewhat ignored [6]. A report released by the United Nations [7] also shows that it is urgently necessary to pay more attention to mental health issues amid the COVID-19 pandemic and that depression and anxiety symptoms have increased in some countries where more investment is urgently needed to prevent a mental health crisis. Therefore, not only does the occurrence and transmission path of the disease itself need attention, but the impact of COVID-19 on public sentiment also needs to be systematically assessed. However, existing studies overly focus on the negative impact of COVID-19 on people’s emotions and are one-sided, ignoring the positive factors that generate positive emotions during the pandemic, e.g., a government’s positive policy responses, the care of medical staff and neighbors, the confidence produced by the improvement of the pandemic, and the motivations of oneself and others. Recent studies on COVID-19 have not sufficiently explored the emotional dimension of group expression; thus, it is necessary to build a quantitative evaluation model of multidimensional emotions.

Twitter [8], Weibo [9], Instagram [10] and other social media platforms have become important channels for people to express their feelings and attitudes. Massive social media data have the properties of openness, communication and extensive public participation—effective bases for capturing public opinion and implementing intervention measures in public events. Social media, as a distributed sensor system [11], can provide a strong foundation for detecting the spatial-temporal characteristics of social public opinion and help improve our cognition and response to public events. Liu, Y. [12] proposed the concept of “social sensing”, whereby the data-mining technology of social media has been widely applied to measure public sentiment when public events occur via, e.g., hot-topic trend analysis [13], in-depth content mining [14], commercial recommendation systems [15], disaster detection [16,17] or spatial-temporal characteristic evolution analysis [18,19]. Crooks, A [11] collected Twitter data following the 5.8-magnitude earthquake on the east coast of the United States, which could be used to measure the identification and localization of the earthquake’s impact range with a high degree of approximation regarding the disaster-area-investigation data disclosed by US authorities. Wang, B. [9] used Weibo data to evaluate the spatial-temporal characteristics of public sentiment in each stage of flood disasters to improve the resilience of cities to natural disasters.

Particularly since the outbreak of COVID-19, relevant neighborhood studies have made new progress globally [20,21,22,23]. Abd-alrazaq, A [13] researched the main topics and top concerns of tweeters during the pandemic period of COVID-19 by collecting a large number of tweets and analyzing word frequency and emotion. Lwin, M. O [24] studied the global trends of fear, anger, sadness and happiness during the COVID-19 pandemic by analyzing over 20 million Twitter posts, finding that global sentiments have shown rapid evolution. Yang, Y. [21] developed a public sentiment analysis model using grounded theory, collecting big data from Weibo to analyze the spatial and temporal distribution of Chinese public panic during the two waves of the COVID-19 pandemic, finding that public sentiment shifted from delayed, negative and impulsive at the beginning to rational. However, few studies have considered the spatial-temporal clustering and evolution characteristics of emotion or analyzed its correlation with the distribution of pandemic cases. Moreover, machine learning methods, such as natural language processing (NLP), are widely used in social data analysis [25]. Based on the emotion dictionary method, text semantic analysis has been conducted to create an emotion dictionary for public health events, which has strong validity and reliability [26]. Furthermore, public sentiment is often the unity of temporality, space and sociality; thus, an analysis of time series, spatial differentiation and evolutionary characteristics can reveal the mechanism of the pandemic’s impact on mental health [27]. In geography, kernel-density analysis [28], local Moran’s I (LISA Statistics) [29], and Getis-Ord Gi* [30] are commonly used to detect the degree of spatial concentration of elements. Additionally, standard deviation ellipse analysis is usually applied to observe the direction of spatial agglomeration. Therefore, it is of strong reliability and pioneering potential to introduce the above methods to analyze the nonlinear evolution and clustering characteristics of public sentiment amidst the pandemic.

In addition, previous studies on the emotional geography of COVID-19 have mostly been conducted at a large spatial scale [21,31,32], e.g., research on the COVID-19 pandemic worldwide [33] or nationwide [34] for panel data analysis. Alternatively, others have focused only on large cities at the epicenter of an outbreak, such as New York [35], London [36] or Wuhan [37], while small cities have been ignored to some extent. Bell, D. [38] emphasizes the importance of studying small cities, which, although small, are perfectly formed. The World Health Organization [39] reported that social equity issues need to be addressed amidst COVID-19 because vulnerable groups are hit harder in terms of infection and mental health. The underdeveloped level of medicine, backward housing environment and shortage of community services in small cities are not conducive to treatment and follow-up rehabilitation for COVID-19 [40]. Therefore, it is significant and urgent to study the public emotional response to the pandemic in small cities. This study mainly focuses on small cities, providing a new research scale for the construction of public sentiment maps under pandemic conditions. Shaoxing, a small city in eastern China, has experienced a typical COVID-19 outbreak in distinct stages. Studying the evolution in the spatial-temporal characteristics of public sentiment in Shaoxing can broaden the scope of COVID-19-related research at the small-city scale. Moreover, we acquired social media data with refined geographic coordinate information, denoised it, and calculated its multidimensional emotional intensity and average value in the 100 m × 100 m-grid emotional envelope. This is beneficial to guide the government toward precise pandemic prevention policies and thus differentiated governance strategies for different communities.

In summary, it is necessary to measure the spatial-temporal characteristics and nonlinear evolution mechanism of public sentiment amid the pandemic in a multidimensional, multiscale and fine manner. Therefore, we propose a spatial-based pandemic-cognition-sentiment (PCS) conceptual model, which helps build a framework to correlate public sentiment with pandemic distribution characteristics and expands the research path on the spatial-temporal pattern of public response. Given the lack of refined research on the evolution of the spatial-temporal characteristics of emotional responses in small cities, our measurement of public emotion is quantified, spatialized, refined and multidimensional. Specifically, we use Shaoxing, a small city in eastern China, as a case study to analyze its spatial-temporal characteristics and evolution before and after a round of COVID-19. Our findings verify our initial hypothesis: there is obvious agglomeration and differentiation in the spatial-temporal pattern of public sentiment, while the spatial-temporal evolution characteristics are closely related to the distribution characteristics of cases. Finally, based on our findings, we propose three operational and targeted public opinion management measures via the perspective of spatial intervention, which can help improve the mental health resilience of residents in response to future pandemics.

## 2. Materials and Methods

### 2.1. Study Area

The study area is Shaoxing city, Zhejiang Province, a small city located in the southeastern coastal area of China, as shown in Figure 1. Shaoxing city is located in the Yangtze River Delta, between latitudes 29°13′35″ to 30°17′30″ north and longitudes 119°53′03″ to 121°13′38″ east. Shaoxing is a prefecture-level city under the jurisdiction of Zhejiang province, covering an area of 8256 square kilometers with a permanent population of 5.27 million as of 2020. Since the COVID-19 pandemic outbreak began in Shaoxing on 8 December 2021, COVID-19 cases have been mainly distributed in Shangyu district and Yuecheng district within Shaoxing city.

### 2.2. Data Sources

The dataset mainly includes the spatial and temporal distribution of confirmed COVID-19 cases in Shaoxing in December 2021 and social media check-in data. The data of administrative boundaries at the district, county and subdistrict levels in Shaoxing city were obtained from the data of zoning codes and urban-rural division codes used by the National Bureau of Statistics in 2016.

The daily numbers and trajectory data of confirmed COVID-19 cases in Shaoxing were mainly collected from the official Wechat public platform “Shaoxing Release”, where the government of Shaoxing released all the pandemic-related information to the public. A Python crawler program was written to obtain the daily numbers and tracking data of confirmed COVID-19 cases in Shaoxing city from 7 December to 31 December 2021. We collected all the confirmed cases that were reported, 387 cases in total; 184 cases were tracked, with a total of 2003 detailed tracks.

The microblog check-in data of Shaoxing city were extracted mainly by Python, and a total of 88,484 COVID-related Weibo data (including the microblog text, geographic location, latitude and longitude, release time, etc.) were obtained. To ensure data quality, the noise-filtering method [41] was applied to denoise the data (excluding Weibo and advertisements forwarded by users and published by robots). Finally, a total of 20,975 microblog check-in data from 8 November to 7 December 2021, and 20,439 microblog check-in data related to the pandemic from 8 December to 31 December 2021 were retained; some of the Weibo data are selected to display in Table 1. Based on these data, we conducted a text-semantic analysis based on NLP.

### 2.3. Pandemic-Cognition-Sentiment (PCS) Conceptual Model

Combining cognitive behavior and emotional geography theory, we propose the pandemic-cognition-sentiment (PCS) conceptual model, which includes pandemic situations, humans, emotions, and spatial behavior, as shown in Figure 2. The pandemic is conducted through ontology and spatial exposure, and the understanding of the pandemic among humans can be gradually deepened via the dangers of pandemic ontology, spatial proximity and outbreak scale at the macro level and of direct and indirect contact infection at the micro level. Subsequently, through the “input” and “information filtering” of cognitive behavior theory, pandemic information is recognized via the influence of sensory acceptance ability and sensibility, and various stable concepts are generated. The human brain expresses these concepts of stability as emotions, which can be described in terms of their nature (positive, neutral, negative), intensity and elasticity. According to the connotation of their emotional geography, public sentiment is directly connected with spatial geography, i.e., the spatial expression of emotions. Spatial pattern analysis is conducted using the four dimensions of sentiment strength, spatial-temporal distribution, emotional vulnerability and resilience. According to their emotions, people decide whether to produce active and passive feedback to the pandemic and whether to express their emotions and produce spatial behaviors (practical activities). If there are spatial-temporal behaviors, then the collection of these spatial behaviors will eventually affect and transform the pandemic environment, allowing researchers to explore the interactive relationship between public sentiment and the pandemic environment.

### 2.4. Methods

#### 2.4.1. Text Semantic Analysis

(1)High-frequency word analysis

Python was used to analyze the high-frequency words in the microblog text to identify the key topics that residents were concerned with during the pandemic. First, the Jieba word segmentation component was used to segment the text, remove the stop words and mark the corresponding parts of speech in the segmentation results. Jieba word segmentation is a common system of Chinese word segmentation based on machine learning [42]. We then used the collection for word frequency statistical analysis to obtain the occurrence frequency of each word in the microblog text. Finally, according to the obtained high-frequency words, Word-Cloud was used to generate a word-cloud map.

(2)Dictionary-based Sentiment Analysis (DSA)

The DSA method was adopted to analyze the sentiment of microblog text and measure the sentiment state and strength reflected by calculating the sentiment scores of each microblog text. We create a sentiment dictionary of public health, which is a BosonNLP dictionary based on data sources such as Weibo, news stories and forums. The final sentiment score is determined by combining a positive and negative word dictionary, degree adverb dictionary and stop word dictionary. The positive and negative scores reflect the nature of emotions, including positive, neutral and negative emotions, and the absolute value of the scores reflects the intensity of sentiment. The flow for calculating a sentiment score in Python is shown in Figure 3. For spatial analysis, we adopt a 100 m × 100 m grid to construct the public sentiment map of Shaoxing and apply the inverse distance weight method (IDW) to process the difference for the regions lacking data. Meanwhile, the effectiveness and accuracy of sentiment recognition were compared with other machine learning algorithms. Through the validation of our test set, it was proven that the method in this experiment is up to 87.63% effective, which meets the international standard.

#### 2.4.2. Spatial Pattern and Correlation Analysis

(1)Kernel-density analysis

The kernel-density analysis tool is used to calculate the density of elements in the surrounding area and the data aggregation of the whole area according to the input factor data, generating a continuous density surface [28]. Its mathematical formula is expressed as follows:(1)$$f_{n}\left(x\right)=\frac{1}{{\mathrm{nh}}_{n}}\overset{n}{\underset{i\;=1}{\sum }}k\left(\frac{x-x_{i}}{h_{n}}\right)$$
where n is the number of points; $h_{n}$ is the bandwidth; $k\left(\frac{x\;-{\;x}_{i}}{h_{n}}\right)$ is the kernel function; x is the position of the point to be estimated; $x_{i}$ is the position of the event point; and k is the spatial weight coefficient. Kernel-density analysis was used to study the spatial distribution pattern of the trajectory points of confirmed cases.

(2)Getis-ord Gi*

Getis-ord Gi* Spatial hot-spot analysis can reflect the distribution of hot spots and cold spots in the local space of the research object by identifying each element in the neighboring element environment [30]. In this paper, cold/hot spots of public emotion are analyzed to characterize the clustering relationship between the relatively low value and high value of public emotion distribution. Local statistics of Getis-ord can be expressed as:(2)$$G_{i}^{*}=\frac{\sum_{j\;=1}^{n}w_{i,j}x_{j}-\;\bar{X}\sum_{j\;=1}^{n}w_{i,j}}{S\sqrt{\frac{\left[n\sum_{j\;=1}^{n}w_{i,j}^{2}-{\left(\sum_{j\;=1}^{n}w_{i,j}\right)}^{2}\right]}{n-1}}}$$
where $x_{j}$ is the attribute value of element j, $w_{i,j}$ is the spatial weight between elements i and j, n is the total number of elements, and $\;\bar{X}=\frac{\sum_{j\;=1}^{n}x_{j}}{n},\;S=\sqrt{\frac{\sum_{j\;=1}^{n}x_{j}^{2}}{n}-{\left(\bar{X}\right)}^{2}}$. The purpose of cold/hot spot analysis is to judge the differentiation characteristics of locus points and the emotional spatial distribution of confirmed cases.

(3)Standard deviational ellipse (SDE)

Standard deviational ellipse (SDE) is a geographical statistical method used to describe the spatial distribution characteristics of geographical elements [43]. In this method, the center, axis and azimuth of the standard deviation ellipse are used to analyze the spatial distribution characteristics of the research object, such as centrality, directivity and spatial morphology. Its main parameters are calculated as follows:

Weighted average center:(3)$${\bar{X}}_{w}=\frac{\sum_{i\;=1}^{n\;}w_{i}x_{i}}{\sum_{i\;=1}^{n}w_{i}}\text{;}{\;\bar{Y}}_{w}=\frac{\sum_{i\;=1}^{n}w_{i}y_{i}}{\sum_{i\;=1}^{n\;}w_{i}}$$

Azimuth of ellipse direction:(4)$$\tan\theta =\frac{\left(\sum_{i\;=1}^{n}w_{i}^{2}{\tilde{x}}_{i}^{2}-\sum_{i\;=2}^{n}w_{i}^{2}{\tilde{y}}_{i}^{2}\right)+\sqrt{{\left(\sum_{i\;=1}^{n}w_{i}^{2}{\tilde{x}}_{i}^{2}-\sum_{i\;=1}^{n}w_{i}^{2}{\tilde{y}}_{i}^{2}\right)}^{2}+4\sum_{i\;=1}^{n}w_{i}^{2}{\tilde{x}}_{i}^{2}{\tilde{y}}_{i}^{2}}}{2\sum_{i=1}^{n}w_{i}^{2}{\tilde{x}}_{i}{\tilde{y}}_{i}}$$

xAxial standard deviation:(5)$$\sigma_{x}=\sqrt{\frac{\sum_{i\;=1}^{n}{\left(w_{i}{\tilde{x}}_{i}\cos\theta -w_{i}{\tilde{y}}_{i}\sin\theta \right)}^{2}}{\sum_{i\;=1}^{n}w_{i}^{2}}}$$

yAxial standard deviation:(6)$$\sigma_{y}=\sqrt{\frac{\sum_{i\;=1}^{n}{\left(w_{i}{\tilde{x}}_{i}\sin\theta -w_{i}{\tilde{y}}_{i}\cos\theta \right)}^{2}}{\sum_{i\;=1}^{n}w_{i}^{2}}}$$
where $\left(x_{i},y_{i}\right)\;$represents the geospatial coordinates of the research object, $w_{i}\;$represents the weight, $\left({\bar{X}}_{w},{\bar{Y}}_{w}\right)\;$represents the weighted average center, θ is the azimuth angle of the ellipse, ${\tilde{x}}_{i}\;$and ${\tilde{y}}_{i}$, respectively, represent the coordinate deviation from the location of each research object to the average center, and $\sigma_{x}\;$and$\;\sigma_{y}$, respectively, represent the standard deviation along the x and along the y axis.

(4)Moran’s I index

Due to the interaction between geospatial data and surrounding areas and the diffusion effect in space, adjacent areas or regions within a certain range from each other are no longer independent but interact with each other. Thus, the data have spatial autocorrelation. Moran’s I index is an important index that is used to measure spatial correlation, and its calculation formula is as follows [44]:(7)$${\mathrm{Moran}}^{\prime }\;I=\frac{n\sum_{i\;=1}^{n}\sum_{j\;=1}^{n}w_{\mathrm{ij}}\left(x_{i}-\bar{x}\right)\left(x_{j}-\bar{x}\right)}{\left(\sum_{i\;=1}^{n\;}\sum_{j\;=1}^{n}w_{\mathrm{ij}}\right)\sum_{i\;=1}^{n}{\left(x_{i}-\bar{x}\right)}^{2}}$$
where $\bar{x}=\frac{1}{n}\sum_{i\;=1}^{n}x_{i}$; $x_{i}$ represents the sample value of the first i region, n is the total number of research units, $w_{\mathrm{ij}}$ and is the spatial adjacency between i regions and j regions, namely, the spatial weight.

Local Moran’s I index specifically displays the spatial correlation of all individuals in a region on the basis of global autocorrelation [29]. The calculation formula of the local Moran’s I index is [44]:(8)$${\mathrm{Moran}}^{\prime }s\;I=\frac{n\left(x_{i}-\bar{x}\right)\sum_{j\;\neq \;i}^{n}W_{\mathrm{ij}}\left(x_{j}-\bar{x}\right)}{\sum_{i\;=1}^{n}{\left(x_{i}-\bar{x}\right)}^{2}}$$
where $\bar{x}=\frac{1}{n}\overset{n}{\underset{i\;=1}{\sum }}x_{i}$; $x_{i}$ represents the sample value of the first i region, n is the total number of research units, and $w_{\mathrm{ij}}\;$is the spatial adjacency between i regions and j regions, namely, the spatial weight.

## 3. Spatial-Temporal Pattern of Confirmed COVID-19 Cases

### 3.1. Overall Spatial-Temporal Distribution of Confirmed Cases

For the spatial dimension, the spatial characteristics of single-core outbreaks in Shaoxing were obvious. The first case was reported in the Shangyu district of Shaoxing city, and subsequent developments showed that the majority of confirmed cases were in the Shangyu district. As of 24:00 on 31 December, Shaoxing had reported 387 confirmed cases, including 384 in Shangyu district and 3 in Yuecheng district, which accounted for 99.22% of the total. The Moran’s I index calculated for the track points of all confirmed cases collected in this round of the pandemic was 6.83, with a confidence degree less than 0.0001, indicating that the track points of confirmed cases had significant clustering characteristics. According to kernel-density analysis (Figure 4a), confirmed cases were mainly concentrated in the Cao E subdistrict and Baiguan subdistrict and their surroundings, forming an obvious cluster. It was spatially restricted to Shangyu district’s central region, indicating that the overall outbreak control was relatively strong.

For the temporal dimension, according to the process and characteristics of the COVID-19 outbreak in Shaoxing (Figure 4b), the pandemic in Shaoxing can be divided into three stages: incubation period, outbreak period and recovery period. Accordingly, we performed follow-up analyses of the incubation period (from 8 November to 7 December 2021) and the outbreak period (from 8 December to 31 December 2021).

### 3.2. Spatial-Temporal Characteristics of the Transmission of Confirmed Cases

The spatial-temporal characteristics of confirmed cases’ distribution varied at different spatial scales. The hot/cold spot analysis (Figure 5a) shows that the hot spots formed on Cao E subdistrict and Baiguan subdistrict in Shangyu district, that the secondary hot spots formed in Yuecheng district, Keqiao district, Zhuji city, Shengzhou city and Xinchang county, and that the other areas were secondary cold spots. The spread of the pandemic is the spread of infection at the microscale, that is, the spread of the disease through media or direct contact within the scope of the traditional transmission channels in the field of medicine. At the medium and macro scales, it is manifested as hierarchical diffusion; that is, from the regional perspective, the pandemic spreads in the direction with less spatial resistance with the help of carriers. According to the spatial characteristics of confirmed cases in Shaoxing, nearly 68% of confirmed cases were concentrated in the Cao E subdistrict, Baiguan subdistrict, Lianghu subdistrict and their surrounding areas in Shangyu district. The characteristics of pandemic spread were obviously spatially adjacent, following the first law of geography. The remaining cases’ tracks did not accord with the distance-attenuation type of gradual spreading and presented a hierarchical diffusion trend. For example, Yuecheng district, Hekeqiao district and the surrounding county and city centers of Shangyu were more significantly affected by the pandemic than the peripheral areas of Shangyu. Infectious diseases need certain people or things as medium carriers because the limitation of space mobile resistance through mobile network transmission is more efficient. Thus, any mobile network that occupies a higher level and establishes more close contact nodes is more vulnerable to the influence of the pandemic.

Meanwhile, Local Moran’s I (Figure 5b) showed that a high-high cluster formed in the Cao E subdistrict, Baiguan subdistrict and Lianghu subdistrict in Shangyu district, indicating that the activity track of confirmed cases was mainly concentrated in the above areas. However, a large contiguous-low cluster formed in southwestern Shaoxing city, indicating that the subdistricts in southwestern Shaoxing city were less affected by the pandemic. Four low-high outliers formed around the high-high clusters, indicating that the tracks of confirmed cases in the four regions were significantly smaller than those in the high-high clusters since human factors interfered with their spatial distribution characteristics. As the actual impact on other subdistricts was relatively small, the COVID-19 pandemic was well controlled and limited to the high-high clusters.

The center of the distribution of confirmed cases in Shaoxing shifted. According to Table 2, the centroid of confirmed cases in the prepandemic incubation period was first located at 120.804 E, 29.985 N. The center of confirmed cases in the outbreak period was located at 120.798 E, 29.965 N, which moved approximately 2.6 km from the previous stage to the southwest by 1.75 degrees. The standard deviation ellipse area in the outbreak period increased by 21.4% compared with that in the incubation period. The results indicate that the activity space of confirmed cases moved westward to the south during the outbreak period, which was reflected in the large-scale infection event caused by a fresh market in the Cao E subdistrict, leading to a shift in the activity space of the pandemic. We also found that the spatial distribution of the activity tracks of confirmed cases in the outbreak period was expanding. To some extent, we believed that the spatial distribution of activity tracks of confirmed cases collected in the outbreak period was more scattered because confirmed patients might have a larger activity space.

## 4. Evolution of the Spatial-Temporal Pattern of Public Sentiment toward COVID-19

### 4.1. Spatial-Temporal Distribution of Public Sentiment before and Amidst COVID-19

The pandemic had a great impact on public sentiment. The intensity of public sentiment fluctuated greatly after the pandemic, and the frequency of both negative and positive words related to the pandemic increased. Based on DSA analysis of high-frequency words (Figure 6a), after the outbreak of the pandemic, not only negative words, such as “pain”, “sadness” and “depression” but also positive semantic words, such as “come on” and “hope”, increased in frequency. Some people thus tended to be depressed after the pandemic, while others tended to post positive motivational statements, including “Come on” and “hope”, on social media to convey emotional encouragement amid the pandemic. Similarly, residents paid great attention to the words “pandemic”, “eating”, “isolation” and “living”, indicating that their daily necessities were the most concerning issue amidst COVID-19. Meanwhile, by analyzing the strength of public sentiment and its daily change (Figure 6b), we find that after the outbreak of the pandemic, the intensity of public emotion showed an overall upward trend with a significantly increased fluctuation range. Hence, the pandemic has made a great difference in people’s emotions, and public sentiment has shown an evolving trend from relatively stable to unstable.

Due to the impact of the COVID-19 pandemic, the public sentiment in Shaoxing showed a trend of multilevel differentiation. In general, the negative mood increased compared with before the pandemic. According to the histogram of the kernel-density distribution of public emotion before and amid the pandemic (Figure 6c,d), the public sentiment before the outbreak mainly ranges from 0 to 50, while the original emotion value of the microblog under the influence of the pandemic mainly ranges from −25 to 25, while the absolute value of both are mainly concentrated at approximately 5. The difference further indicates that the pandemic has a certain negative impact on residents’ emotions.

The spatial-temporal distribution of public sentiment showed obvious differences before and after the outbreak of COVID-19, especially in the pandemic center and its surroundings. According to the public sentiment map before and after the outbreak (Figure 7), the public sentiment in Shaoxing was generally positive before the outbreak. Although Shangyu district’s mood was relatively depressed, it still showed a positive tendency. Shopping- and entertainment-related texts had a high degree of positive sentiment, such as posts at E-tour town and Wanda Plaza. The sentiment map after the COVID-19 outbreak implied that the sentiment in Shangyu district, the epicenter of the outbreak, was generally higher than that of surrounding districts. Through observing posts of relatively high values, we found that positive sentiment mainly came from motivational response words, such as “refueling”, “hope” and “expectation”, reflecting people’s encouragement to each other and confidence in overcoming the crisis during the pandemic. Public sentiment formed a relatively high value aggregation area in the Baiguan subdistrict and Cao E subdistrict of Shangyu district, while a “negative sentiment zone” was generated around the outbreak area of Shangyu (Figure 7d). Through further analysis of the causes through Weibo data, we found that the negative values mainly come from “depression”, “depression” and “discomfort” caused by “isolation”, “nucleic acid testing” and other pandemic control policies that impeded residents’ behaviors.

### 4.2. Evolution of Public Sentiment Characteristics before and Amidst COVID-19

The characteristics of public sentiment response and intensity agglomeration before and after the COVID-19 outbreak in Shaoxing showed some differences at different spatial scales. According to the clustering and outlier analysis through Local Moran’s I (Figure 7c,f), the low-high cluster formed on Baiguan subdistrict, Cao E subdistrict and Dongguan subdistrict in Shangyu district was significantly reduced. Further analysis of Weibo texts showed that the reason for the change in the outlier value was that after the outbreak, the frequency of positive and motivational words increased more than that of negative words, which resulted in an increase in emotional value instead of a decrease.

The distribution center of the intensity of public sentiment shifted to the northeast in Shaoxing, and the degree of spatial agglomeration increased. The emotional response of residents in Shangyu district to the pandemic was more prominent than that in surrounding areas. According to Table 3, the intensity center of public sentiment during the incubation period was located at 120.527 E, 29.902 N; the intensity center of public sentiment shifted to 120.686 E and 29.974 N during the outbreak period. The space centroid offset was 17.291366 km, and the offset angle was 62°27′49.02″. As Figure 8b shows, the short axis of the ellipse was significantly shortened, and the spatial distribution of the represented emotion values had an increased degree of agglomeration, with more obvious direction distribution characteristics. The ellipse moved to the northeast as a whole, and the centroid shifted to Shangyu district, the epicenter of the pandemic, indicating that the number of microblog posts and emotional intensity in Shangyu district increased, and residents in Shangyu district had a stronger emotional response to the pandemic.

At the subdistrict scale, the intensity of the public sentiment response to the pandemic showed a trend of clustering in spatial distribution, and the clustering direction was related to the distribution of confirmed cases. The standard deviation ellipse analysis of sentiment intensity distribution was performed in Baiguan and Cao E subdistricts of Shangyu district, where the COVID-19 pandemic was most severe. The results (Figure 8b) showed that the two standard deviation ellipses tended to be more clustered, and both of them clustered toward the communities with more confirmed cases, indicating that the distribution of confirmed cases had a significant correlation with public sentiment.

## 5. Discussion

The above results reveal the nonlinear and clustering characteristics of the spatial-temporal pattern of public sentiment response to COVID-19 and that public sentiment is closely correlated with the spatial-temporal trajectory distribution of confirmed cases. Our findings regarding three major evolution characteristics are summarized as follows:

(1) **The sentiment fluctuation of residents in the center of the outbreak is obvious.** By comparing the evolution of public sentiment and the trajectory distribution of confirmed cases before and after the outbreak of COVID-19, we found that the public sentiment value of the three streets with the most serious outbreak of the epidemic changed from negative to positive, and the sentiment value of this cluster significantly increased. In terms of the overall trend, the fluctuation and upward trend of emotion value were the most obvious in the regions where the activity trajectories of confirmed cases were concentrated. Further analysis of specific microblog content implied that the rise of emotional value in the postoutbreak stage was mainly due to the stimulation of encouraging texts, indicating that residents in this stage had entered the encouragement period from the anxiety and tension period earlier, and artificial active emotional adjustment had occurred.

(2) **The areas around the pandemic cluster formed a “low sentiment zone” with increased negativity.** Compared with the pandemic period, the sentiment intensity in the subdistricts around the areas with high infection rates decreased to a certain extent, especially in the streets of Yuecheng district. In addition, a “low emotional pressure zone” with relatively low sentiment value formed around the pandemic clusters. The negative emotions in the microblog content mainly came from the conflict between strict prevention and control measures and residents’ necessary activities rather than the impact of pneumonia symptoms themselves. Therefore, it can be concluded that under the joint prevention and control mechanism of the epidemic, the government should not only pay attention to the mental health of the people in epidemic areas but also pay attention to the emotional changes of residents in the nonoutbreak core areas and reduce the impact of authorities’ restrictions on residents’ travel behaviors.

(3) **The sentiment of the central subdistricts surrounding the outbreak fluctuated significantly and showed a downward trend.** By overlaying the evolution of public sentiment and the spatial distribution of the confirmed cases before and after the outbreak, it can be found that the central streets of Yuecheng district, Zhuji city, Shengzhou city and Xinchang county around Shangyu district had a more obvious decline in sentiment value. This phenomenon is consistent with the characteristics of epidemic transmission at the regional scale; that is, regions with closer links to the epidemic center and lower travel costs have higher epidemic risk values. The public sentiment in the surrounding areas with a dense distribution of confirmed cases was also vulnerable to shocks and fluctuations, indicating that the impact of COVID-19 on mood fluctuations was also influenced by spatial proximity and accessibility.

As mentioned above, the spread of COVID-19 has a great impact on public sentiment and even mental health; this impact is nonlinear and related to control policies and community organizations. Therefore, we propose targeted strategies for future pandemic conditions via the three dimensions of constructing psychological defense lines, joint prevention and control mechanisms, and social livelihood guarantees from the perspective of spatial intervention. The specific strategies are as follows:

(1) **Build a comprehensive mental health intervention system.** Psychological intervention should respond to pandemic events at any time, move the time threshold forward, and intervene in advance through education in pandemic prevention measures, especially for the center of a pandemic outbreak and its surrounding areas [45]. The inconvenience and uncertainty under the pandemic risk and social isolation caused by the strict control measures around the epicenter of the outbreak were the key factors of negative sentiment that need to be considered and addressed. It is necessary to establish professional psychological working groups among the multiple scales of city, district and subdistrict in a timely manner to partially reduce violations of pandemic prevention regulations and curb the spread of negative emotions among residents.

(2) **Jointly release early warning maps for the spread of COVID-19.** The direct infection risk of pandemic exposure is not transmitted uniformly but in a gradient manner according to the grade of high-centrality nodes in an urban network. Therefore, it is necessary to timely update the pandemic risk map according to available epidemiological survey data. Warning for the spread of the epidemic should be issued for nearby areas with high centrality and close connections to the surrounding network. Additionally, the existing and potential risk sources should be identified as early as possible.

(3) **Construct block-level resilient urban living space.** Weibo posters paid more attention to words such as “epidemic”, “eat” and “life” after the outbreak. According to Maslow’s theory of needs, security and basic physiological needs are the lowest level of needs and the source of people’s most primitive survival anxiety, suggesting that we should pay attention to the resilience construction of residents’ living space. Therefore, the creation of 15-min lifecycles is suggested, that cater to the basic needs of residents with more public green space. They can be used for daily leisure in peacetime and provide space for emergency facilities in wartime. At the same time, the government should invest more in the construction of health stations, supermarkets and other basic infrastructure to form a resilient urban living space with balanced resource distribution. In this way, the residents’ worries about basic living security can be eliminated to some extent, and the psychological fluctuation of residents during the outbreak can be alleviated.

## 6. Conclusions

As there is a lack of fine research at the small-city scale, we proposed a spatial-based pandemic-cognition-sentiment (PCS) conceptual model and conducted a case study of Shaoxing city, China, at a small-city scale. Massive media data were collected to calculate the public sentiment. The methods of natural language processing (NLP), Getis-Ord Gi*, and standard deviation ellipse, etc., were used to measure the spatial-temporal characteristics of confirmed cases and the evolution pattern of public sentiment under the influence of COVID-19. Our findings verify that there is obvious agglomeration and differentiation in the spatial-temporal pattern of public sentiment, while the spatial-temporal evolution characteristics are closely related to the distribution characteristics of cases after the outbreak. Finally, targeted spatial strategies are proposed to improve people’s mental health resilience under pandemic conditions. The main conclusions are as follows:

(1) **The spread of the pandemic is characterized by contagion and diffusion at the micro level and hierarchical diffusion at the macro level.** According to the spatial characteristics of confirmed cases, the pandemic spread through droplet or contact transmission at the microscale, which reflects the characteristics of infection spread. At the macro scale, viruses spread in the direction of least resistance to spatial diffusion. With the help of the mobile network and its carriers, it preferentially propagates to the nodes with high centrality and high adjacency in the urban flow network, mainly showing the spatial characteristics of hierarchical diffusion.

(2) **The pandemic has suppressed the emotional impact of residents in the center of the outbreak cluster.** The sentiment value of the outbreak center increased significantly in the later stage of the outbreak, and this increase in sentiment positivity was mainly due to the stimulus of encouraging words. It is shown that with the development of the epidemic, public sentiment in the outbreak clusters gradually changed from low to proactive motivation and that there was a certain degree of emotional repression–incentive response.

(3) **The pandemic has a negative gradient impact on public sentiment in the surrounding area.** The COVID-19 pandemic had a significant negative impact on the emotions of residents in the areas surrounding the outbreak clusters. The intensity of the impact showed a gradient decline in terms of spatial proximity and accessibility, reflecting that the residents in the areas closer to the outbreak center were more significantly affected by the pandemic. Therefore, it can be concluded that the negative impact of the pandemic on public sentiment in surrounding areas with higher centrality and greater accessibility in the urban flow network is more obvious to a certain extent.

However, there are still shortcomings in this study. In terms of data representativeness, the user groups of Weibo have certain particularities, and the blog data that we collected may not represent the views of people of all ages. Additionally, there is room for further improvement in the accuracy of the sentiment analysis. Moreover, factors affecting residents’ emotions are complex, including their physical and social environments, and their stressors include disaster events, personal stressors, background stressors, etc. It is possible to further study the mechanism of how these factors affect public emotions. In future research, the spatial-temporal evolution characteristics of public sentiment analyzed in this paper could be introduced into analyses regarding the prediction and management of public sentiment in response to future pandemics. Adopting targeted spatial interventions, as mentioned above, or timely soft measures, such as positive social media bots [46], could help boost residents’ optimism and increase mental health resilience amid a crisis. Additionally, the methodology of our research can be further applied to many other fields, such as accurate disaster detection and spatial-temporal characteristic evolution analysis of public sentiment. The category of disaster not only includes pandemics, but also earthquakes, floods, tornados and any other disasters. The results can provide reference for the dynamic adjustment of emergency disaster decision making and urban planning strategies, so as to enhance the resilience and sustainability of cities.

## Figures and Tables

**Figure 1 ijerph-19-11306-f001:**
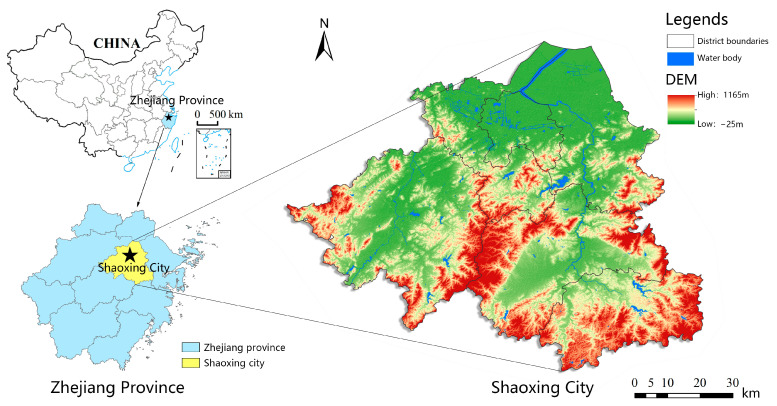
Study Area.

**Figure 2 ijerph-19-11306-f002:**
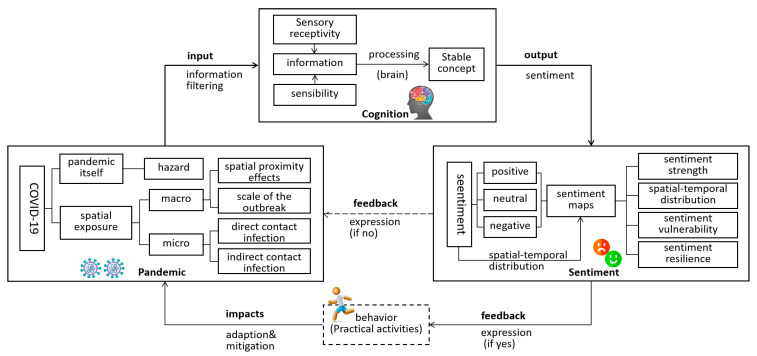
Pandemic-cognition-sentiment (PCS) conceptual model.

**Figure 3 ijerph-19-11306-f003:**
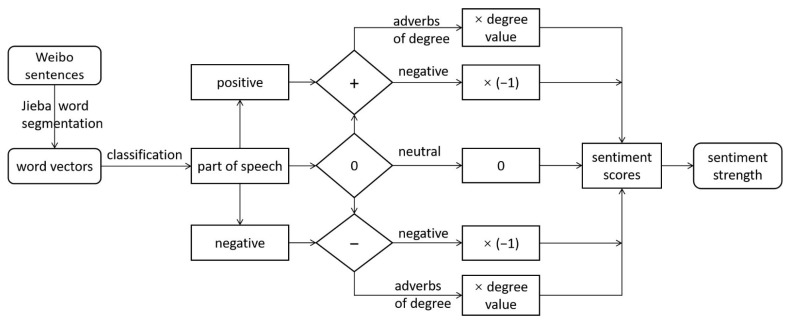
Dictionary-based sentiment analysis (DSA).

**Figure 4 ijerph-19-11306-f004:**
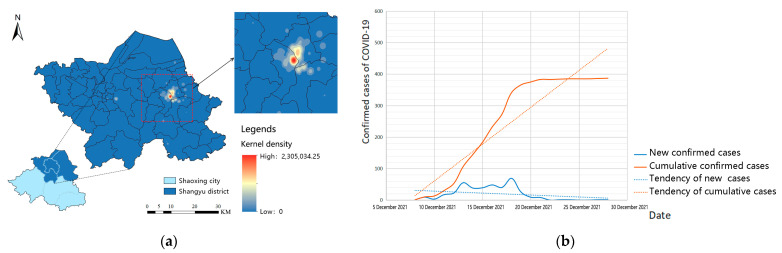
Spatial-temporal characteristics of confirmed COVID-19 cases in Shaoxing city: (**a**) Kernel-density analysis of confirmed COVID-19 cases; (**b**) Line chart of daily number of new and cumulative confirmed cases.

**Figure 5 ijerph-19-11306-f005:**
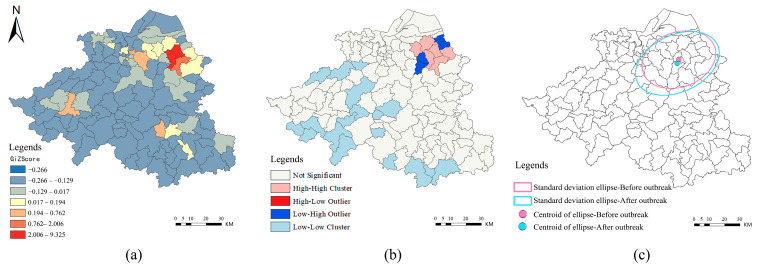
Analysis of the spatial-temporal characteristics of the distribution of confirmed COVID-19 cases in Shaoxing: (**a**) Gi-ZScore hot-spot analysis; (**b**) Clustering and outlier analysis—local Moran’s I, and (**c**) Standard deviation ellipse analysis.

**Figure 6 ijerph-19-11306-f006:**
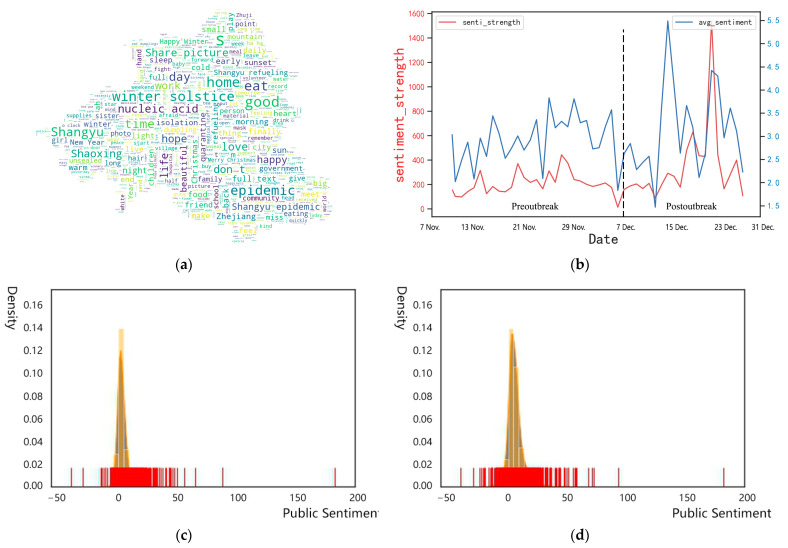
Analysis of the word cloud and kernel-density distribution of microblogs before and during the pandemic: (**a**) Word cloud map of pandemic-related microblogs in Shaoxing. (**b**) Line chart of daily public sentiment strength before and after the pandemic. (**c**) Histogram of kernel-density distribution of prepandemic public sentiment. (**d**) Histogram of kernel-density distribution of public sentiment after the pandemic outbreak.

**Figure 7 ijerph-19-11306-f007:**
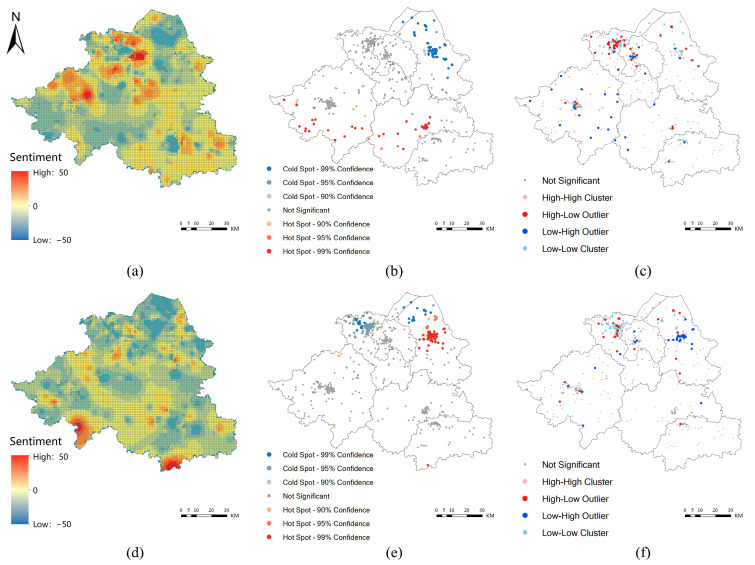
Analysis of public sentiment, cold/hot spots and local Moran’s I before and amidst COVID-19: (**a**) Prepandemic public sentiment map; (**b**) Prepandemic cold-hot spots; (**c**) Prepandemic local Moran’s I; (**d**) Postoutbreak public sentiment map; (**e**) Postoutbreak cold/hot spot analysis; (**f**) Postoutbreak local Moran’s I.

**Figure 8 ijerph-19-11306-f008:**
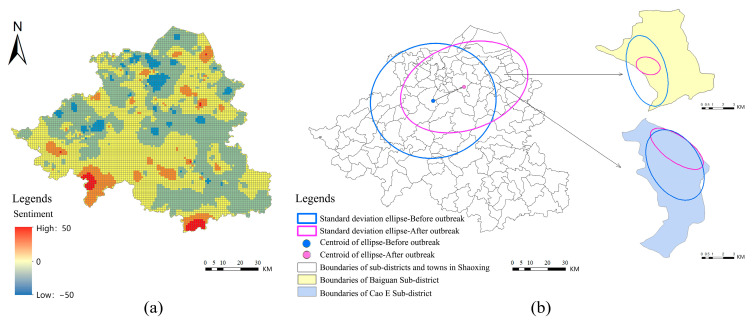
Spatial-temporal evolution of public sentiment before and after the COVID-19 outbreak. (**a**) Difference in public sentiment before and after the outbreak: (**b**) Transformation of the standard deviation ellipse during the outbreak. (Shaoxing City, Baiguan subdistrict and Cao E subdistrict).

**Table 1 ijerph-19-11306-t001:** Selected Weibo Blog Data Display.

ID	Weibo Content *	Release Time	Location *
1	Running to isolation points in high-risk areas every day, dealing with contact personnel every day, having endless police calls every day, sleeping less than two hours a day...	Monday, 13 December 21:45:07	Shaoxing Keqiao Wanda Plaza
2	“Zhejiang world has so many people”. Volunteers Shangyu refueling day 8, we are the best!	Saturday, 18 December 19:32:15	Wolong Tian Xiang Hua Ting
3	As soon as I came out to see such a beautiful sky, to my anxiety was suddenly added a gleam of joy.	Tuesday, 21 December 17:11:03	School of Science and Art, Zhejiang Sci-tech University
4	25 December—On the fifth day after entering the isolation ward, I didn’t want to say anything more, but just wanted to go home. I was really under great pressure at work. I didn’t sleep except for a few hours every day, and I was anxious the rest of the time.	Saturday, 25 December 05:18:59	Shaoxing Municipal Hospital (Central Hospital)
5	So many people are refueling for Shangyu, so many people are supporting us. Come on, hang in there, hang in there!	Saturday, 25 December 18:57:51	Shangyu e travel town

* Weibo content is translated from Chinese. Location includes latitude and longitude.

**Table 2 ijerph-19-11306-t002:** Comparison of characteristic data of the standard deviation ellipse of the activity trajectory of confirmed cases during the incubation period and outbreak period.

Period	Shape_Length	Shape_Area	Center_X	Center_Y	XStdDist	YStdDist	Rotation
Before	124,630.134	1,209,491,384	120.804 E	29.985 N	22,136.22	17,392.96	58.698
During	139,247.056	1,468,638,980	120.798 E	29.965 N	25,949.98	18,015.85	56.950

**Table 3 ijerph-19-11306-t003:** Comparison of standard deviation ellipse characteristics of public sentiment distribution in Shaoxing city before and during the outbreak period.

Period	Shape_Length	Shape_Area	Center_X	Center_Y	XStdDist	YStdDist	Rotation
Before	201,018.265	3,210,865,478	120.527 E	29.902 N	32,511.53	29,348.01	79.432
During	180,985.136	2,505,983,588	120.686 E	29.974 N	33,234.58	24,002.90	64.708

## Data Availability

The data in this study are available in the article.

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
