# Peer review of "Spatial-Temporal Pattern Evolution of Public Sentiment Responses to the COVID-19 Pandemic in Small Cities of China: A Case Study Based on Social Media Data Analysis"

_ijerph, 2022, doi:10.3390/ijerph191811306_

Round 1

Reviewer 1 Report

The article is interesting; however, in the reviewer's opinion, it is worth comparing the results obtained in the authors' study with analogous studies for other cities/countries.

It would also be advisable to describe more extensively the selection of source data - especially the number of diagnosed COVID-19 cases. To what extent is the selection of data by the method described representative?

Author Response

Thanks for your comments and suggestions! Please see the attachment. 

Reviewer 2 Report

It is another article about the impact of the COVID-19 pandemic on public mental health. Many authors have studied it. 

However, it should be noted that the reviewed text innovatively explores the spatial-temporal characteristics of public sentiment responses to COVID-19 exposure to improve urban anti-pandemic decision-making and public health resilience. It uses modern research methods such as Local Moran's I, kernel density analysis, 17 Getis-Ord Gi* and standard deviation ellipse.

Paper is well organized. Abstract reflects the article content. Introduction presents state of knowledge and the objective of the research. Paper presents the research methodology, result and discussion. The author carefully presents results of the research and analysis. At the end author presents conclusions and references. 

For the article I report the following miner comments:
- lines 11-12, I can not agree with the statement “…as most recent studies have focused on the macro scale or large cities, there is a lack of fine research at the small city scale”. For example, please see the article: Majewska A. et al., 2022, Pandemic resilient cities: Possibilities of repairing Polish towns and cities during COVID-19 pandemic. Land Use Policy 113, https://doi.org/10.1016/j.landusepol.2021.105904

-Figure 6a), word cloud map of pandemic-related microblogs in Shaoxing should be translated to English

-Figures 4-5 and 7 - legends are not very readable, to be corrected

-lines 341-342 Table 2 title should be moved to the next page

Author Response

(The authors gave the same response as above.)

Reviewer 3 Report

It is an inspiring and meaningful article.  I have only a few recommendations for the authors to consider.  They include:

1. Even though the authors propose a pandemic-cognition-sentiment conceptual framework for the study, the authors do not specify the expected results.  It is not easy for me to relate the results to the framework.

2. The methodology is very inspiring.  The authors should propose any further application of the method.

Author Response

(The authors gave the same response as above.)
